# Preliminary Evidence of the Potent and Selective Adenosine A2B Receptor Antagonist PSB-603 in Reducing Obesity and Some of Its Associated Metabolic Disorders in Mice

**DOI:** 10.3390/ijms232113439

**Published:** 2022-11-03

**Authors:** Magdalena Kotańska, Anna Dziubina, Małgorzata Szafarz, Kamil Mika, Marek Bednarski, Noemi Nicosia, Ahmed Temirak, Christa E. Müller, Katarzyna Kieć-Kononowicz

**Affiliations:** 1Department of Pharmacological Screening, Jagiellonian University Medical College, 9 Medyczna Street, PL 30-688 Krakow, Poland; 2Department of Pharmacodynamics, Jagiellonian University Medical College, 9 Medyczna Street, PL 30-688 Krakow, Poland; 3Department of Pharmacokinetics and Physical Pharmacy, Jagiellonian University Medical College, Medyczna 9, PL 30-688 Cracow, Poland; 4Division of Neuroscience, Vita Salute San Raffaele University, 20132 Milan, Italy; 5PharmaCenter Bonn, Pharmaceutical Institute, Pharmaceutical & Medicinal Chemistry, An der Immenburg 4, D-53121 Bonn, Germany; 6Chair of Technology and Biotechnology of Drugs, Faculty of Pharmacy, Jagiellonian University Medical College, Medyczna 9, PL 30-688 Cracow, Poland

**Keywords:** adenosine A_2B_ receptor antagonist, glucose tolerance, metabolic disorder, obesity, PSB-603, Theophylline

## Abstract

The adenosine A_2A_ and A_2B_ receptors are promising therapeutic targets in the treatment of obesity and diabetes since the agonists and antagonists of these receptors have the potential to positively affect metabolic disorders. The present study investigated the link between body weight reduction, glucose homeostasis, and anti-inflammatory activity induced by a highly potent and specific adenosine A_2B_ receptor antagonist, compound PSB-603. Mice were fed a high-fat diet for 14 weeks, and after 12 weeks, they were treated for 14 days intraperitoneally with the test compound. The A_1_/A_2A_/A_2B_ receptor antagonist theophylline was used as a reference. Following two weeks of treatment, different biochemical parameters were determined, including total cholesterol, triglycerides, glucose, TNF-α, and IL-6 blood levels, as well as glucose and insulin tolerance. To avoid false positive results, mouse locomotor and spontaneous activities were assessed. Both theophylline and PSB-603 significantly reduced body weight in obese mice. Both compounds had no effects on glucose levels in the obese state; however, PSB-603, contrary to theophylline, significantly reduced triglycerides and total cholesterol blood levels. Thus, our observations showed that selective A_2B_ adenosine receptor blockade has a more favourable effect on the lipid profile than nonselective inhibition.

## 1. Introduction

In a large number of obese patients, low to severe inflammation associated with white adipose tissue can be observed. Subsequent activation of the immune system leads to insulin resistance, glucose intolerance, and diabetes. In obesity, white adipose tissue produces inflammatory agents, including tumour necrosis factor-alpha (TNF-α) and interleukin-6 (IL-6), which not only have local effects on the physiology of adipose tissue but additionally induce systemic effects in other organs [1,2,3]. Increased levels of TNF-α and IL-6, in addition to many other factors, are the cause of insulin resistance. Furthermore, IL-6 increases the production of C-reactive protein (CRP) in the liver and can indirectly promote the appearance of cardiovascular disorders [1]. Inflammation also affects β islet cells and alters insulin secretion [4].

Promising therapeutic targets in the treatment of obesity and diabetes are the adenosine receptor subtypes A_2A_ and A_2B_ [5,6,7,8]. The agonists [7] and antagonists [9] of A_2A_ adenosine receptors and the agonists and antagonists of A_2B_ adenosine receptors have the potential to positively affect metabolic disorders [7]. Adenosine A_2A_ receptor agonists can induce the anti-inflammatory action necessary for the survival of β cells and increase insulin secretion [7], reduce food intake [10], and increase thermogenesis and lipolysis [6]. Adenosine A_2A_ receptor antagonists can exert anti-inflammatory effects and reduce IL-6 levels [9]. The activation of the A_2B_ adenosine receptor can reduce the dysfunction of β cells, lead to a reduction in hyperglycaemia and insulin resistance and exert anti-inflammatory effects in various cells including adipocytes [7,11]. Antagonists of the A_2B_ adenosine receptor may cause effects, such as increased insulin release [7,8], reduced insulin resistance, decreased fat accumulation in the liver [7], reduced liver glucose production, improved glucose disposal into skeletal muscles and brown adipose tissue in diabetic mice [12], inhibition of the progression of renal fibrosis derived from diabetes [13], induction of anti-inflammatory effects [7,14], and reduction of IL-6 levels [14,15]. Furthermore, obesity is associated with colon inflammation, leading to an increased A_2B_ receptor expression; as a result, the A_2B_ receptor can mediate motor dysfunctions of the colon frequently seen in obesity [16].

Since A_2A_ and A_2B_ adenosine receptors regulate glucose homeostasis in diabetes and obesity ligands of these receptors may be useful for the prevention and treatment of obesity-associated metabolic disorders.

Theophylline is a standard adenosine receptor antagonist, and it blocks non-selectively all subtypes of this receptor in humans [17,18]. This natural plant alkaloid exhibits micromolar affinities at adenosine receptors [18]. The K_i_ value of theophylline at the A_2B_ adenosine receptor, depending on the species, equals 9070–74,000 nM; 15,100 nM, and 5630 nM for humans [19,20], rat [20], and mice [21] receptors respectively. Some studies demonstrated a beneficial effect of theophylline on body weight in obese animals [22,23], but its side effects, such as hyperactivity [24], and heart rhythm disturbances [25,26] turned out to be a serious problem. It is thus of interest to examine whether a more selective antagonist of the A_2B_ adenosine receptor would also be effective in reducing body weight, but at the same time be safer than chronically applied theophylline.

The present study investigated the link between glucose homeostasis and anti-inflammatory activity induced by a blockade of the A_2B_ adenosine receptor using a highly potent and selective adenosine A_2B_ receptor antagonist [27,28]. The compound PSB-603 (Figure 1) displays > 17,000-fold selectivity over other adenosine receptors (K_i_ values are 0.553 > 100,000 > 10,000 > 10,000 nM for human A_2B_, A_1_, A_2A_, and A_3_ receptors, respectively), and is similarly potent and selective in rats and mice [27,28]. The K_i_ value of PSB-603 at the A_2B_ adenosine receptor, depending on the species, equals 0.553 nM; 0.355 nM, and 0.265 nM for human-, rat-, or mice receptors, respectively [27]. Thus, PSB-603 acts much stronger at the A_2B_ adenosine receptor than theophylline.

We have previously studied its effect on the inflammatory process in models of inflammation caused by the administration of zymosan A or carrageenan, and its antioxidant activity in vitro [14]. In this study, we present the results of the effects of PSB-603 on body weight in an obesity model caused by the administration of high-fat feed and sucrose, as well as on the amount of peritoneal fat, cholesterol and triglyceride levels, glucose and insulin tolerance, and spontaneous activity in mice.

## 2. Results

### 2.1. Effect of PSB-603 Administration on Body Weight

Mice fed a high-fat diet and a sucrose solution showed greater weight gain during the 12-week period of inducing obesity than control animals. Animals in the control group fed the standard diet weighed approximately 30 g, while mice in the obesity control group fed a high-fat diet weighed approximately 38 g—26.6% more. Animals fed a high-fat diet and treated with PSB-603 (for 14 days) at a dose of 2 × 5 mg/kg body weight (b.w.)/day showed significantly less weight gain than mice in the obese control group (F(52,494) = 5.236, *p* < 0.0001). From the 12th day of compound administration, a statistically significant difference in body weight was observed between these groups (Figure 2A). PSB-603 reversed the effect of the high-fat diet and significantly reduced body weight (F(4,39) = 14.23, *p* < 0.0001) (Figure 2B). Theophylline, which served as a positive control, had a much weaker effect. The results are shown in Figure 2A,B.

### 2.2. Effect of PSB-603 Administration on Peritoneal Fat Pads

The group of animals, which received the tested compound at a dose of 2 × 5 mg/kg b.w., had a significantly lower amount of fat in the peritoneum compared to obese control mice (F(4,34) = 16.49, *p* < 0.0001). In the other two tested groups treated with theophylline or PSB-603 at a dose of 5 mg/kg b.w./day, no significant decrease in body fat was found compared to the control obese group. The results are shown in Figure 3.

### 2.3. Effect of PSB-603 Administration on Locomotor and Spontaneous Activity of Mice

Locomotor activity was determined in mice after a single intraperitoneal administration of PSB-603 at doses of 10, 5, and 1 mg/kg b.w. Only at a dose of 10 mg/kg b.w. did the antagonist significantly decrease locomotor activity (F(3,30) = 9.743, *p* = 0.0001). The results are shown in Figure 4.

Spontaneous activity was determined in obese mice. A single administration of PSB-603 at a dose of 5 mg/kg b.w. and its subchronic (13 days) administration resulted in significant reductions in spontaneous activities compared to the control groups in both phases of the day during the first and/or third hours of observation (the light phase), and in the fifth, seventh, eighth, tenth, or thirteenth hours of observation (the dark phase). The results are shown in Figure 5A,B.

After administration of PSB-603 at a dose of 2 × 5 mg/kg b.w. (9.00 am and 1.00 pm) only in the first hour after the second administration mice showed reduced mobility compared to the control group. On the thirteenth day of the experiment, the activity of mice treated with PSB-603 (2 × 5 mg/kg) was very close to the activity of mice from the control group, which received only solvent (1% Tween 80). Only in the first hour after the second administration was the activity higher (Figure 5A,B).

Theophylline (2 × 50 mg/kg) significantly increased the activity of mice in the light phase on the first day and after subchronic (13 days) administration. The results are shown in Figure 5A,B.

### 2.4. Effect of PSB-603 Administration on the Plasma Glucose Levels

There were no significant differences in plasma glucose levels between all groups of obese mice (Figure 6) indicating that neither PSB-603 nor theophylline affected the increased plasma glucose levels in these mice.

### 2.5. Glucose Tolerance and Insulin Sensitivity Tests

Fasting glucose concentrations were measured after 20 h of food deprivation just before the glucose loading test. Blood glucose levels in obese control mice were significantly higher at all time points compared to the blood glucose levels observed in control mice (fed standard feed). On the contrary, blood glucose levels in mice fed a high-fat diet and treated with PSB-603, measured at the same time points, did not differ significantly from the levels determined in control mice fed a standard diet. Furthermore, blood glucose levels at 30 and 60 min after glucose load in all treated groups were significantly lower than glucose levels in obese control mice (F(12,102) = 6.483, *p* < 0.0001) (Figure 7A). As shown in Figure 7B, the area under the curve (AUC) decreased after PSB-603 (at a dose of 2 × 5 mg/kg b.w.) or theophylline treatment compared to the obese control group.

The initial glucose levels before the insulin sensitivity test were significantly lower in the PSB-603 treated group (2 × 5 mg/kg) compared to the levels in the obese control group (Figure 7C). Similar results were observed for theophylline. In the insulin test, there were no statistically significant differences in the glucose levels between the tested groups, AUC values were also not statistically different (Figure 7D).

### 2.6. Triglyceride and Total Cholesterol Levels

The level of triglycerides in the blood was higher in obese mice than in mice fed a standard diet. Animals treated for two weeks with PSB-603 at a dose of 2 × 5 mg/kg b.w./day and fed a high-fat diet had a significantly lower plasma triglyceride level compared to the control group fed a high-fat diet (F(4,44) = 6.672, *p* = 0.0003). There was no statistically significant difference between both groups treated with PSB-603 and fed a high-fat diet and the group fed standard feed. The level of triglycerides in the group treated with theophylline was comparable to the level in the obese control group. However, the SD in this group was relatively high. Results are shown in Figure 8A.

Total cholesterol levels in all obese groups were higher than in the standard-feed control group. There were no significant differences in total plasma cholesterol levels between the obese control group and the PSB-603 treated group at a dose of 2 × 5 mg/kg b.w./day. In contrast, total cholesterol was significantly lower in the groups treated with PSB-603 at a dose of 5 mg/kg b.w. or with theophylline compared to the control group fed high-fat feed (F(4,44) = 23.28, *p* < 0.0001) (Figure 8B).

### 2.7. TNF-a and IL-6 Levels

In the obese control mice, plasma levels of IL-6 and TNF-α were higher than in the standard-fed control mice. However, the TNF-α level was significantly reduced by PSB-603 administration for 14 days at doses of 5 mg/kg and 2 × 5 mg/kg b.w./day (F(4,45) = 4.292, *p* = 0.005). In the group treated with theophylline, similar changes were observed (Figure 9A). The level of IL-6 was significantly reduced by PSB-603 administration for 14 days only at the dose of 2 × 5 mg/kg b.w./day (F(4,38) = 3.415, *p* = 0.005) as well as by theophylline (Figure 9B).

## 3. Discussion

The purpose of this study was to investigate whether a selective A_2B_ adenosine receptor antagonist, PSB-603, can significantly affect body weight and selected biochemical parameters related to obesity as well as inflammatory processes that occur in obese animals. Will chronic administration of PSB-603 lead to weight reduction? Will the potential positive effects be fraught with undesirable side effects, such as increased locomotor activity seen with the non-selective theophylline? An established compound, the selective A_2B_ adenosine receptor antagonist PSB-603, which is similarly potent and selective in humans, mice, and rats [27], was chosen to answer these questions.

The study began with experiments to determine the compound’s effect on locomotor activity in CD-1 mice after its single administration at doses of 10, 5, or 1 mg/kg b.w. This was a screening study conducted to select the appropriate dose for chronic experiments. Changes in mobility, such as sedation or excessive stimulation, could alter or falsify the effects of the tested compound on body weight [29,30]. Only at a single dose of 10 mg/kg b.w. PSB-603 significantly reduced the activity of the mice. For this reason, a dose of 5 mg/kg b.w. (for which the anti-inflammatory effect of this compound had previously been demonstrated [9]) and a dose of 2 × 5 mg/kg b.w. were selected for chronic studies. A study by Pardo et al. showed that even a single administration of theophylline at doses of 5–15 mg/kg b.w. can potentiate locomotor activity [31]. Nevertheless, in our study, the reference compound theophylline was administered at a dose of 2 × 50 mg/kg b.w. in accordance with the literature data [32]. PSB-603 administrated to obese mice at a dose of 2 × 5 mg/kg b.w. did not have a significant effect on spontaneous activity, which was demonstrated by the chronic mobility monitoring using the telemetry method. The results of the weight reduction are therefore certainly not due to sedation, where animals often eat less. In contrast, monitoring the activity of obese mice treated with theophylline confirmed its stimulating effect at the tested dose.

In a subsequent study, we evaluated the effects of the potent and selective xanthine-based antagonist PSB-603 [28] and the methylxanthine derivative theophylline, a non-selective adenosine receptor antagonist that blocks all four receptor subtypes (A_1_, A_2A_, A_2B_, and A_3_) in humans, and three subtypes in rodents (A_1_, A_2A_, A_2B_) [18], on body weight in mice with diet-induced (high-fat/sucrose) obesity. The results obtained by us clearly show that theophylline administered at a dose of 2 × 50 mg/kg and PSB-603 administered at a dose of 5 mg/kg or 2 × 5 mg/kg significantly reduced body weight in obese mice. Our results are different from those obtained by Gnad et al., who showed that activation, but not blockade of the A_2B_ receptor, protected against obesity induced by a high-fat diet (HFD) [33]. In turn, Tofovic et al., 2016, showed that combined blockade of A_1_ and A_2B_ adenosine receptors significantly increased body weight, but still had beneficial effects on multiple biochemical markers associated with obesity.

Adenosine A_2B_ receptors are located in key tissues related to obesity, i.e., adipocytes or liver and skeletal muscles, where they play an important regulatory role [33,34]. They are also detected in human and animal adipose tissue [34] and their expression is increased in abdominal adipose tissue of mice fed a high-fat diet [35]. Therefore, animals with the missing A_2B_ gene (A_2B_ receptor KO mice) gain more weight than wild type mice, have an increased accumulation of visceral adipose tissue, and developed insulin resistance in an HFD model [36,37]. In vitro studies showed the ability of this receptor to inhibit adipogenesis and lipogenesis [38]. Although A_2B_ receptors are known to be involved in the regulation of body weight, it is still not known whether the reduction of body weight and the decrease in the amount of adipose tissue are due to their stimulation or blockade. The absence of a definitive answer to this question may be due in part to the different models of obesity used by researchers and the use of different time courses of activation or inhibition of the A_2B_ receptor. Genetic deletion of the A_2B_ receptor in adipose tissue of mice fed a high-fat diet is known to result in increased weight gain [37]. However, it is not known whether chronic blockade may lead to its up-regulation, and thus cause a different effect than it seems directly logical (lack of receptors or blockade of receptors—weight gain). Therefore, we suggest that research that explains the potential influences of multiple administrations of selective ligands on changes in the densities of receptors located in individual tissues, which are important in the pathogenesis of obesity, as well as the correlation with the effect on body weight, should be carried out in the future.

Subsequently, our research showed that loss of body weight was accompanied by a reduction in the amount of peritoneal fat. This effect was statistically significant after repeated administration of the selective adenosine A_2B_ receptor antagonist PSB-603. Anti-obesity effects of non-selective methylxanthines blocking A_1_/A_2B_ adenosine receptors have been well described [22]. These compounds can effectively reduce body fat and weight gain induced by high-fat diets in rodents [23], stimulate lipolysis, and inhibit adipogenesis through several molecular mechanisms [22,39]. However, in our study, theophylline, as a non-selective antagonist of the A_1_/A_2B_ adenosine receptors (as well as the A_2A_ receptor subtype), induced a weaker effect in reducing the amount of adipose tissue compared to the more potent and selective A_2B_ receptor antagonist.

Mice fed a high-fat/sugar diet gained body weight and developed metabolic changes, including disturbances in carbohydrate metabolism and diabetes, as well as hyperlipidaemia [40]. The A_2B_ receptor subtype appears to play a pivotal role in modulating glucose homeostasis and insulin resistance, thus emerging as a promising target for new drugs aiming at the treatment of metabolic disorders [41]. Adenosine stimulates liver glucose production by activating A_2B_ receptors [8,42,43], and selective A_2B_ receptor antagonists induce hypoglycaemia in diabetic mice [44]. In the present study, mice fed an HFD for 14 weeks developed a marked increase in body weight, followed by a marked alteration of several metabolic indices, such as an increase in blood glucose, cholesterol, and triglyceride levels, and an increase in blood levels of the inflammatory markers TNF-α and IL-6. Taking this into account, in the next stage, we attempted to assess whether a selective A_2B_ adenosine receptor antagonist could lower elevated glucose levels and prevent the development of glucose tolerance and insulin resistance. Our observations show that PSB-603 and theophylline did not affect basal glucose levels in the obese state. On the contrary, the selective partial A_2B_ receptor agonist BAY60-6583 significantly reduced plasma glucose levels in mice fed the HFD [35]. However, PSB-603 and theophylline improved glucose tolerance, which was clearly observed in the glucose loading test in our study, and in agreement with the observation that BAY60-6583 decreased insulin resistance [35].

The beneficial effects of PSB-603, a selective A_2B_ receptor antagonist, on body weight and metabolic disturbances, may be mediated by its anti-inflammatory activity and the reduction in plasma levels of IL-6 and TNF-α. In current and previous studies, we demonstrated a high anti-inflammatory potential of PSB-603 [14] as well as another adenosine A_2A_/A_2B_ receptor antagonist (compound KD-64) [9]. In support of our findings, Johnston-Cox et al. showed that expression of the A_2B_ receptor in macrophages plays an important role in controlling insulin sensitivity and glucose tolerance by regulating inflammatory cytokines that affect insulin signalling [45]. Furthermore, they demonstrated that A_2B_ receptor signalling in macrophages plays an important role in the protective effects of the A_2B_ receptor on HFD-induced insulin resistance. The results of Figler et al., suggest that A_2B_ receptor blockade may become an effective way to counteract insulin resistance by altering liver glucose production and reducing the production of IL-6 and other cytokines [46]. Moreover, the authors showed that deletion of the A_2B_ receptor gene and selective blockade of the A_2B_ receptor in mice reduced liver glucose production and increased glucose utilization in skeletal muscles and brown adipose tissue. Other studies showed that the adenosine A_2B_ receptor may modulate whole-body glucose metabolism by regulating insulin receptor substrate-2 (IRS-2) levels. A_2B_ knockout mice had decreased IRS-2 levels, which was associated with impaired insulin signalling in tissues [35].

In our study, chronic feeding of a high fat/sugar diet resulted in higher levels of triglycerides and plasma cholesterol, which is consistent with published results [47,48]. Theophylline, a non-selective antagonist of adenosine receptors, significantly reduced only total cholesterol levels without affecting elevated triglycerides in obese mice. The study by Tovofic et al., showed a beneficial effect of dual A_1_/A_2B_ adenosine receptor blockade on lipid homeostasis, resulting in reduced plasma triglycerides and slightly decreased plasma cholesterol levels [49]. PSB-603, on the other hand, significantly reduced triglycerides and total cholesterol levels (only at the lower daily dose). PSB-603 at a dose that has a more beneficial effect on body weight (2 × 5 mg/kg b.w./day) did not affect the total cholesterol level, which may perhaps be due to an increase in the HDL fraction by this compound, although this issue requires more detailed studies. Thus, we showed that selective A_2B_ receptor blockade has a more favourable effect on the lipid profile than a non-selective blockade. However, the precise role of the adenosine A_2B_ receptor in the regulation of lipid homeostasis is complicated. According to the literature, (i) the A_2B_ receptor is associated with the regulation of liver lipid metabolism, and liver expression of A_2B_ receptors in mice was greatly increased by a high-fat diet [35,50]; (ii) the lack of A_2B_ receptors may have adverse effects on plasma lipids, i.e., it led to the development of liver steatosis with elevated liver and plasma triglyceride and cholesterol levels in HFD-fed mice [50], or it may protect mice from the accumulation of liver triglycerides [51]; (iii) antagonism of the A_2B_ receptor increased the expression of genes required for fatty acid metabolism, while activation of the A_2B_ receptor enhanced lipid accumulation in a cultured mouse hepatocyte cell line [51]. These interesting, often contradictory, results lead to the conclusion that more elaborate studies are needed to elucidate the precise effects of subtype-specific adenosine receptor ligands as well as non-selective compounds, on the lipid profile in specific models of metabolic disorders. The data available in the literature stem from various models that are difficult to compare.

Finally, we want to highlight the possible involvement of more targets, in addition to adenosine receptors, in the activity of individual ligands. Small ligands with a methylxanthine structure, such as theophylline, can significantly affect, for example, the activity of phosphodiesterases [52], and this mechanism may contribute to their actions. Therefore, the full spectrum of activities of a tested compound is usually not known, which we consider a significant limitation of our study, as well as of many published studies.

Other limitations of this study need to be highlighted. Since our research is preliminary, there was only a selected number of tests performed to showcase the potential differences between the activities of investigated compounds and selective and non-selective antagonists of adenosine receptors. The promising results open the field for future, more detailed, investigations which allow showing specific changes induced by PSB-603 administration (by histological and molecular characterization of liver morphological and functional changes during treatment, histological and immunohistochemistry analysis performed at the adipose tissue level, etc). Moreover, a comparison of agonist versus antagonist efficacy in the same metabolic disorder model could explain many of the confusing facts about the actions of adenosine A_2B_ receptor agonists and antagonists in metabolic disorders.

In conclusion, the results obtained by us clearly show that theophylline (non-selective adenosine receptor antagonist) and PSB-603 (selective adenosine A_2B_ receptor antagonist) significantly reduced body weight in obese mice. However, theophylline significantly increased the spontaneous activity of mice which could alter the results. Loss of body weight was accompanied by a reduction in the amount of peritoneal fat; however, this effect was statistically significant only after PSB-603 administration. Theophylline and PSB-603 had no effect on glucose levels in the obese state, but PSB-603, contrary to theophylline, significantly reduced triglycerides and total cholesterol blood levels. Thus, our observations show that selective A_2B_ receptor blockade has a more favourable effect on the amount of peritoneal fat and the lipid profile than a non-selective blockade. However, the exact sequence of molecular events in the organism, connecting influence of PSB-603 on adenosine A_2B_ receptor with weight reduction and improvement of metabolic disturbances, remains an open question and requires further studies.

## 4. Materials and Methods

### 4.1. Animals

In the study, adult female Albino Swiss mice, CD-1, weighing 18–22 g were used. The animals were kept in environmentally controlled rooms, in standard cages lit with artificial light for 12 h each day. The animals had free access to food and water, except for the time of the acute experiment. The randomly established experimental groups consisted of 8–10 mice. All animal care and experimental procedures were carried out in accordance with the European Union and Polish legislation acts concerning animal experimentation and were approved by the Local Ethics Committee of Jagiellonian University in Cracow, Poland (Permission No: 256/2015, 55/2017).

### 4.2. Drugs, Chemical Reagents, and Other Materials

Theophylline was purchased from Sigma-Aldrich (Warszawa, Poland). The compound PSB-603 (Figure 1) was synthesised at PharmaCenter Bonn, Pharmaceutical Institute, Bonn, Germany, according to a described procedure [28]. The identity and purity of the final product were assessed by NMR and LC-UV/MS techniques.

For studies, PSB-603 (5 mg/kg b.w. of the mouse or 2 × 5 mg/kg b.w. of the mouse) was suspended in 1% Tween 80 and the volume was adjusted to 10 mL/kg. This dose was chosen because PSB-603 at 5 mg/kg b.w. does not have a sedative effect (locomotor activity study—Section 2.3) and has anti-inflammatory activity [14]. Theophylline was administered intraperitoneally at a dose of 2 × 50 mg/kg b.w. of the mouse [32].

### 4.3. Experiment Methods

#### 4.3.1. Metabolic Disorders Induced by a High-Fat/Sucrose Diet and Influence of the Tested Compounds on Body Weight and Spontaneous Activity

Mice were fed a high-fat diet consisting of a 40% fat blend (Labofeed B with 40% lard, Morawski Feed Manufacturer, Żurawia, Poland) for 14 weeks, water and a 30% sucrose solution were available ad libitum [53,54]. Control mice were fed a standard diet (Labofeed B, Morawski Feed Manufacturer, Poland) and drank only water. After 12 weeks, mice with diet-induced obesity were randomly divided into four equal groups that had the same mean body weight and were treated intraperitoneally with test compounds at the following doses: PSB-603 5 mg/kg b.w./day; PSB-603 5 mg/kg b.w./two times a day; theophylline 50 mg/kg b.w./two times a day or vehicle—1% Tween 80, 0.35 mL (fat/sugar diet + vehicle = obesity control group) once a day in the morning, between 9:00 and 10:00 am or twice a day at 9:00 am and 2:00 pm for 14 days. Control mice (control without obesity) were kept on a standard diet, with the intraperitoneal administration of vehicle—1% Tween 80, 0.35 mL (standard diet + vehicle = control group). Water and sucrose were measured daily, immediately prior to the morning drug administration. Animals always had free access to food, water, and sucrose.

High-fat feeding composition (932 g of dry mass): protein—193 g, fat (lard)—408 g, fibre—28.1 g, crude ash—43.6 g, calcium—9.43 g, phosphorus—5.99 g, sodium—1.76 g, sugar—76 g, magnesium—1.72 g, potassium—7.62 g, manganese—48.7 mg, iodine—0.216 mg, copper—10.8 mg, iron—125 mg, zinc—61.3 mg, cobalt—0.253 mg, selenium—0.304 mg, vitamin A—15,000 units, vitamin D3—1000 units, vitamin E—95.3 mg, vitamin K3—3.0 mg, vitamin B1—8.06 mg, vitamin B2—6.47 mg, vitamin B12—0.051 mg, folic acid—2.05 mg, nicotinic acid—73.8 mg, pantothenic acid—19.4 mg, choline—1578 mg.

The high-fat diet contained 550 kcal and the standard diet 280 kcal per 100 g.

The spontaneous activity was measured on the first and 13th day of treatment with a special RFID system—TraffiCage (TSE-Systems, Bad Homburg, Germany). Animals were subcutaneously implanted with transmitter identification (RFID), allowing the presence and time spent in different areas of the cage to be counted, and then the data were grouped in a special computer program [55].

#### 4.3.2. Glucose Tolerance Test

The test was performed at the beginning of week 15. After the fourteenth (once daily) or twenty-eighth (twice daily) administration of the test compound, food and sucrose were discontinued for 20 h and then glucose tolerance was tested. Glucose (1 g/kg b.w.) was administered intraperitoneally. Blood samples were taken at time points: 0 (before glucose administration), 30, 60, and 120 min from the tail vein. Glucose levels were measured with a glucometer (ContourTS, Bayer, Leverkusen, Germany, test stripes: ContourTS, Ascensia Diabetes Care Poland Sp. z o.o., Warszawa, Poland, REF:84239666). The area under the curve (AUC) was calculated using the trapezoid rule.

#### 4.3.3. Insulin Sensitivity Test

Insulin tolerance was tested the next day after the glucose tolerance test (after this test, mice had free access to standard food and water). Three hours before the insulin tolerance test, the food was discontinued. Insulin (0.5 IU/kg b.w.) was injected intraperitoneally, blood samples were taken at time points: 0, 15, and 30 min from the tail vein, and glucose levels were measured with a glucometer (ContourTS, Bayer, Leverkusen, Germany, test stripes: ContourTS, Ascensia Diabetes Care Poland Sp. z o.o., Warszawa, Poland, REF:84239666). The AUC was calculated using the trapezoid rule.

#### 4.3.4. Locomotor Activity

Locomotor activity was recorded with an Opto M3 multichannel activity monitor (MultiDevice Software v1.3, Columbus Instruments, Columbus, OH, USA). It was evaluated as the distance travelled by animals when trying to climb [55]. Mice immediately after intraperitoneal administration of the test compound were placed in parameter counting cages; however, the activity measurement was read 30 min after administration of the test compound for a period of 20 min.

#### 4.3.5. Biochemical Analysis

Blood and fat pads were collected after decapitation and then blood was centrifuged at 600× *g* (15 min, 4 °C) to obtain plasma. To determine cholesterol and triglyceride levels in plasma, standard enzymatic spectrophotometric tests (Biomaxima S.A. Lublin, Poland, catalogue number: 1-023-0400 or 1-053-0400) were used. The absorbance was measured at a wavelength of 500 nm.

To determine IL-6 and TNF-α levels LANCE^®^ Ultra Detection Kits (PerkinElmer, Inc., Waltham, MA, USA, catalogue numbers: TRF1505, TRF1504C/TRF1504M) were used. LANCE^®^ Ultra is a homogeneous (no wash) time-resolved fluorescence resonance energy transfer technology.

### 4.4. Statistical Analysis

Statistical calculations were performed using the GraphPad Prism 6 program (Graph-Pad Software, San Diego, CA, USA). The results are presented as arithmetic means with a standard deviation (means ± SD). The normality of the data sets was determined using the Shapiro–Wilk test. Statistical significance was calculated using one-way ANOVA, Tukey post hoc test (two control groups), with significance level set at 0.05 (triglyceride, cholesterol, glucose, IL-6, TNF-α levels, amount of fat pads) or one-way ANOVA, Bonferroni post hoc (one control group), with a significance level set at 0.05 (locomotor activity) or two-way ANOVA, Tukey post hoc test with the significance level set at 0.05 (changes in body weight, glucose tolerance test, insulin tolerance test) or the multiple t test, with significance level set at 0.05 (spontaneous activity). Differences were considered statistically significant at: * *p* ≤ 0.05, ** *p* ≤ 0.01, *** *p* ≤ 0.001.

## Figures and Tables

**Figure 1 ijms-23-13439-f001:**
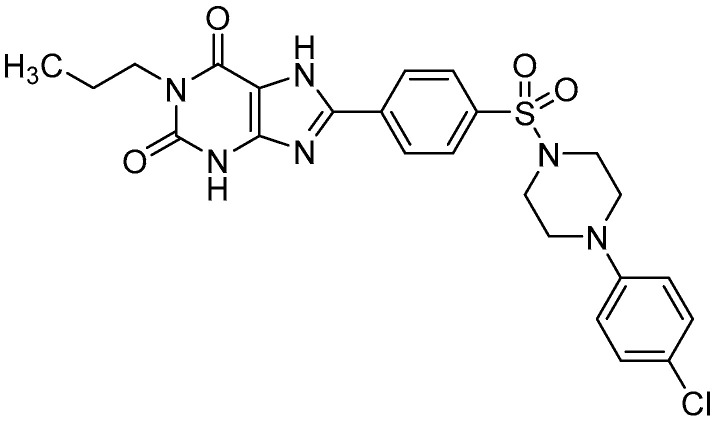
Structure of the potent and selective A_2B_ adenosine receptor antagonist PSB-603.

**Figure 2 ijms-23-13439-f002:**
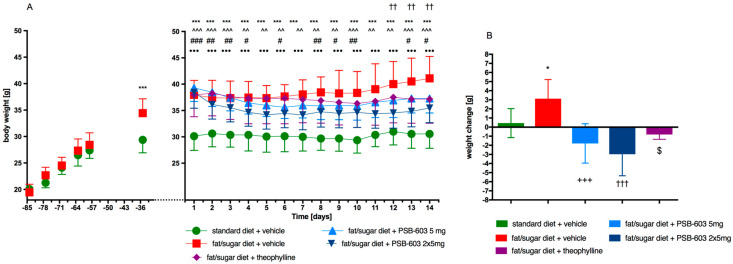
Effect of PSB-603 administration on body weight. (**A**) Body weight throughout the administration. (**B**) Sum of weight changes. Results are expressed as means ± SD, *n* = 8–10. Multiple comparisons were performed by two-way ANOVA, Tukey post hoc, (**A**) or one-way ANOVA Tukey post hoc (**B**). * Significant between control mice fed a standard diet and control mice fed a fat/sugar diet; ^ Significant between control mice fed a standard diet and mice treated with PSB-603 at a dose of 5 mg/kg b.w./day and fed a fat/sugar diet; # significant between control mice fed a standard diet and mice treated with PSB-603 at a dose of 2 × 5 mg/kg b.w./day and fed a fat/sugar diet; • significant between control mice fed a standard diet and mice treated with theophylline at a dose of 2 × 50 mg/kg b.w./day and fed a fat/sugar diet; + significant between control mice fed a fat/sugar diet and mice treated with PSB-603 at a dose of 5 mg/kg b.w./day and fed a fat/sugar diet; † significant between control mice fed a fat/sugar diet and mice treated with PSB-603 at a dose of 2 × 5 mg/kg b.w./day and fed a fat/sugar diet; $ significant between control mice fed a fat/sugar diet and mice treated with theophylline at a dose of 2 × 50 mg/kg b.w./day and fed a fat/sugar diet; *, #, $ *p* < 0.05, ^^, ##, †† *p* < 0.01, ***, ^^^, +++, †††, •••, ### *p* < 0.001.

**Figure 3 ijms-23-13439-f003:**
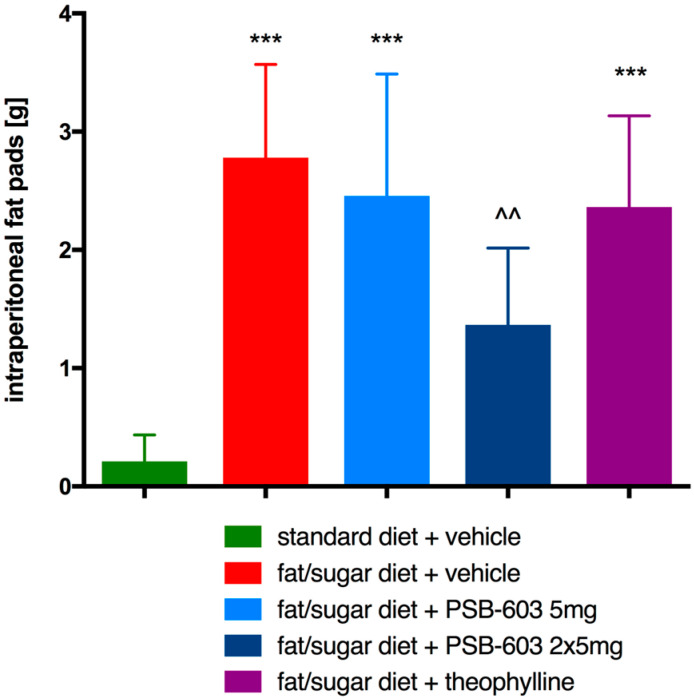
Effect of PSB-603 administration on the mass of adipose pads. Results are expressed as means ± SD, *n* = 7–8. Multiple comparisons were made using Tukey post hoc one-way ANOVA. * Significant against control mice fed standard diet; ^ significant against control mice fed a fat/sugar diet; ^^ *p* < 0.01, *** *p* < 0.001.

**Figure 4 ijms-23-13439-f004:**
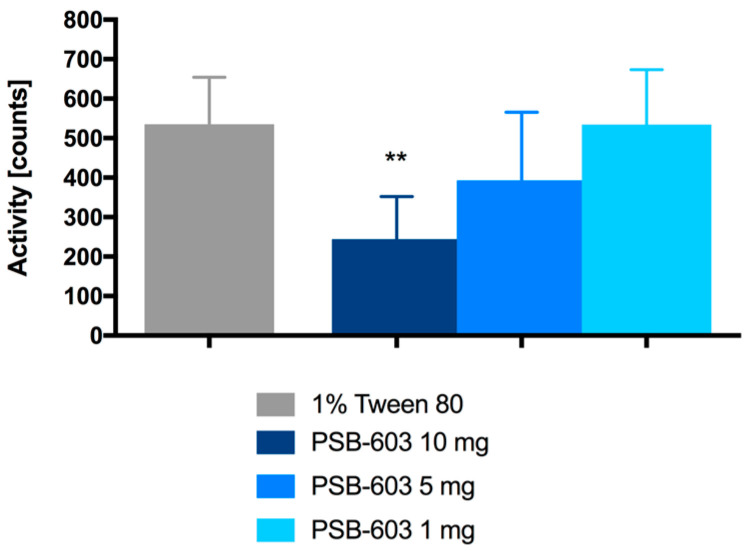
Locomotor activity after a single intraperitoneal administration of PSB-603. Results are expressed as means ± SD, *n* = 8–10. Comparisons were made by one-way ANOVA Bonferroni post hoc, * significant against control mice; ** *p* < 0.01.

**Figure 5 ijms-23-13439-f005:**
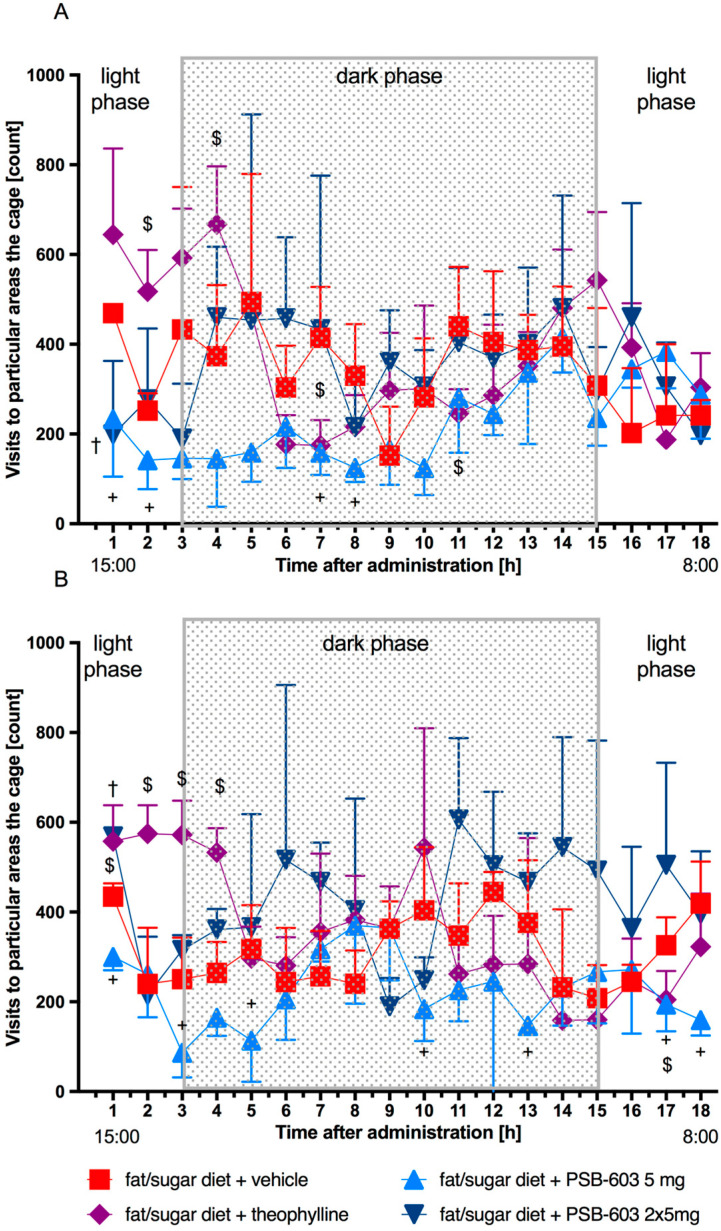
Spontaneous activity measured on the 1st and 13th days of treatment. Results are expressed as means ± SD, *n* = 8. Comparisons were made using the multiple t test; (**A**) after the first dose; (**B**) after 13 days of treatment; + significant between control mice fed a fat/sugar diet and mice treated with PSB-603 at a dose of 5 mg/kg b.w./day and fed a fat/sugar diet; † significant between control mice fed a fat/sugar diet and mice treated with PSB-603 at a dose of 2 × 5 mg/kg b.w./day and fed a fat/sugar diet; $ significant between control mice fed a fat/sugar diet and mice treated with theophylline at a dose of 2 × 50 mg/kg b.w./day and fed a fat/sugar diet; +, †, $ *p* < 0.05.

**Figure 6 ijms-23-13439-f006:**
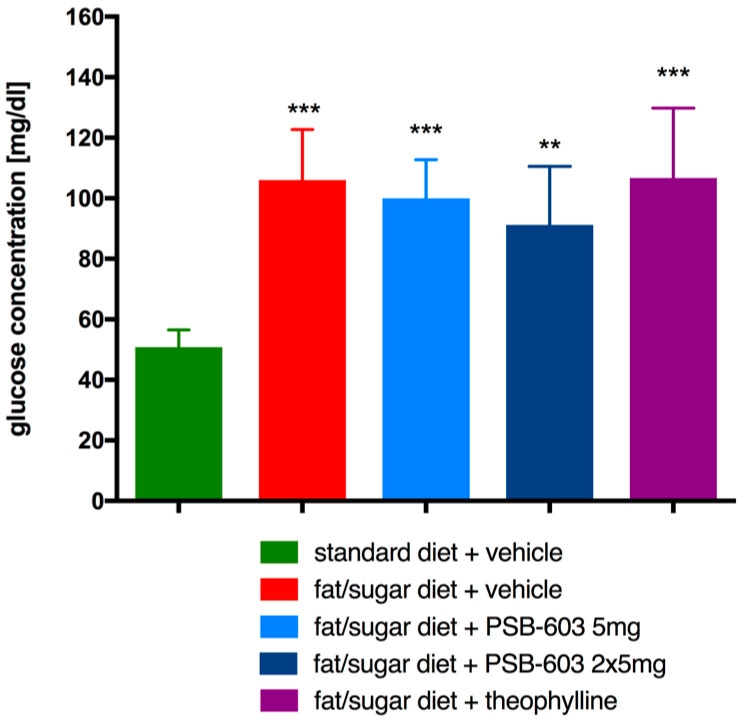
Effect of PSB-603 or theophylline administration on the plasma glucose level. Results are expressed as means ± SD, *n* = 6–9. Comparisons were performed by one-way ANOVA Tukey post hoc; * significant compared to control mice fed standard diet; ** *p* < 0.01, *** *p* < 0.001.

**Figure 7 ijms-23-13439-f007:**
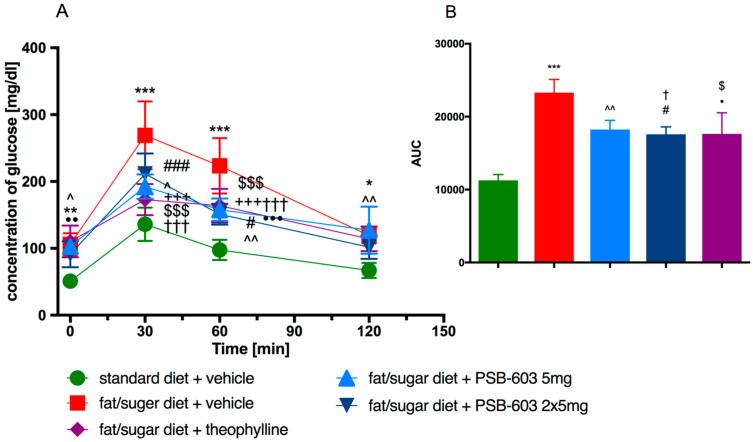
Intraperitoneal glucose tolerance test (IPGTT) (**A**). The area under the IPGTT curve (**B**). Insulin sensitivity test (IST) (**C**). The area under the IST curve (**D**). Results are expressed as means ± SD, *n* = 6–9. Comparisons were made by two-way ANOVA, Bonferroni post hoc; * significant between control mice fed a standard diet and control mice fed a fat/sugar diet; ^ significant between control mice fed a standard diet and mice treated with PSB-603 at a dose of 5 mg/kg b.w./day and fed a fat/sugar diet; # significant between control mice fed a standard diet and mice treated with PSB-603 at a dose of 2 × 5 mg/kg b.w./day and fed a fat/sugar diet; • significant between control mice fed a standard diet and mice treated with theophylline at a dose of 2 × 50 mg/kg b.w./day and fed a fat/sugar diet; + significant between control mice fed a fat/sugar diet and mice treated with PSB-603 at a dose of 5 mg/kg b.w./day and fed a fat/sugar diet; † significant between control mice fed a fat/sugar diet and mice treated with PSB-603 at a dose of 2 × 5 mg/kg b.w./day and fed a fat/sugar diet; $ significant between control mice fed a fat/sugar diet and mice treated with theophylline at a dose of 2 × 50 mg/kg b.w./day and fed a fat/sugar diet; *, #, ^, $, †, • *p* < 0.05, **,••, ^^ *p* < 0.01, ***, +++, †††, •••, ###, $$$ *p* < 0.001.

**Figure 8 ijms-23-13439-f008:**
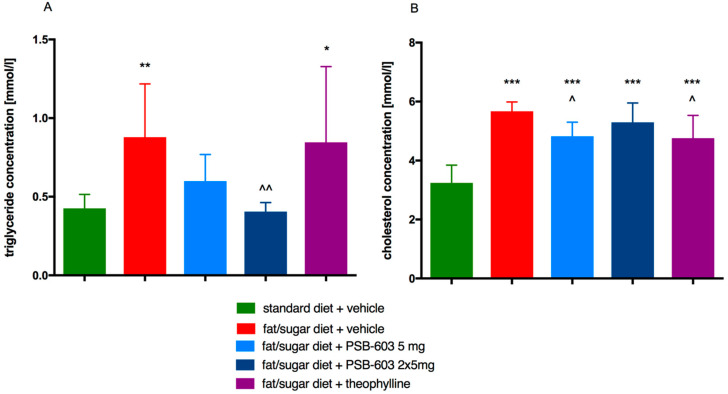
Effect of PSB-603 or theophylline administration on plasma levels of triglycerides (**A**) or total cholesterol (**B**). Results are expressed as means ± SD, *n* = 9–10. Comparisons were performed by one-way ANOVA Tukey post hoc; * Significant compared to control mice fed a standard diet; ^ Significant compared to control mice fed fat/sugar diet; *, ^ *p* < 0.05, **, ^^ *p* < 0.01, *** *p* < 0.001.

**Figure 9 ijms-23-13439-f009:**
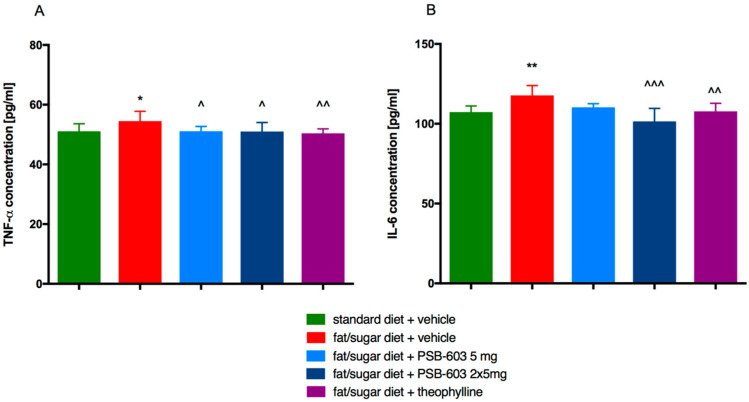
Effect of PSB-603 administration on (**A**) TNF-α and (**B**) IL-6 levels in plasma. Results are expressed as means ± SD, *n* = 8–10. Comparisons were made by one-way ANOVA Tukey post hoc; * significant compared to control mice fed a standard diet; ^ significant compared to control mice fed a fat/sugar diet; *, ^ *p* < 0.05, **, ^^ *p* < 0.01, ^^^ *p* < 0.001.

## Data Availability

The data presented in this study are available upon request from the corresponding author.

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
