# Peer review of "Preliminary Evidence of the Potent and Selective Adenosine A2B Receptor Antagonist PSB-603 in Reducing Obesity and Some of Its Associated Metabolic Disorders in Mice"

_ijms, 2022, doi:10.3390/ijms232113439_

Round 1

Reviewer 1 Report (Previous Reviewer 1)

This is an resubmitted manuscript. The authors have revised the article according the reviewer's comments in the previous review. I have no further comment. 

Author Response

Reviewer 2 Report (New Reviewer)

This manuscript by Kotańska et al. investigated the effect of PSB-603, a selective adenosine A2B receptor antagonist, in obesity model mice showing that PSB-603 treatment could reduce the body weight as well blood TG levels. PSB-603 also improved glucose tolerance. As the authors say, these findings are preliminary evidence that PSB-603 may have a favorable effect on obesity. However, some issues require clarification and re-evaluation. These are all detailed in the comments below:

Major

1.     Why were only female mice used in the experiments? Are there any known sex differences in the adenosine A2B receptor signaling or drugs modulating its function?

2.     Please show the body weight curve starting from the initial body weight before starting HFD feeding. It would also be better if the authors showed the food intake. Even if the activity is not changed, it doesn’t necessarily mean the food intake doesn’t change.

3.     Are the blood glucose concentrations shown in Figure 6 the fasting blood glucose levels? If so, the duration of fasting should be stated. 

4.     Glucose tolerance assessed by intraperitoneal GTT seems to be improved by the treatment of PSB-603 without affecting insulin sensitivity (Figure 7). Then did the authors measure glucose-stimulated insulin secretion? What do the authors think is why the glucose tolerance was improved by PSB-603 treatment? It may be the decrease in the inflammatory cytokine IL-6, but if so, the insulin resistance should be improved.

5.     Adenosine A2B receptor of which organ do the authors think is primarily responsible for the anti-obesity effect of this drug? The authors only show the phenomena but do not determine any mechanisms. There are controversial reports on whether the activation or the blockade of this receptor signaling combats obesity. I think it would be better if the authors showed some mechanistic experiments.

Author Response

This manuscript is a resubmission of an earlier submission. The following is a list of the peer review reports and author responses from that submission.

Round 1

Reviewer 1 Report

The current study investigated bioactivity of the adenosine A2B receptor antagonist, compound PSB-603, on regulating body weight, glucose homeostasis and anti-inflammatory activity using a high-fat diet mouse model. The results suggested that this compound could reduce body weight and blood triglycerides and total cholesterol levels. The introduction provides sufficient background for readers to understand the type and the bioactivity of these adenosine receptors. The experiment is well designed and the discussion is well analyzed to explain the conflicts among several researches. There are two points need to be explained by the authors.

1. At a single dose of 10 mg/kg b.w. PSB-603 significantly reduced the activity of the mice. For what reason that the administration of 2x5 mg/kg b.w. PSB-603 did not show to reduce the activity of the mice since the total among of PSB-603 administered to the mice is equal to 10 mg/kg b.w. a day.    

2. What is the Ki value of theophylline for A2B adenosine receptors? Concerning the potency, can it be compared between 2x50 mg/kg b.w. of theophylline and 2x5 mg/kg b.w. of PSB-603? 

Reviewer 2 Report

Manuscript ID: ijms-1840964

Title: The potent and selective adenosine A2B receptor antagonist PSB-603 reduces obesity and some of its associated metabolic disorders in mice

Authors: Magdalena KotaÅ„ska, Anna Dziubina, MaÅ‚gorzata Szafarz, Kamil Mika, Marek Bednarski, Noemi Nicosia, Ahmed Temirak, Christa Müller, Katarzyna Kieć-Kononowicz

Submitted to section: Molecular Pharmacology

Special Issue: Natural and Synthetic Compounds for Management, Prevention and Treatment of Obesity 2.0

The manuscript ijms-1840964 by Magdalena Kotańska and co-workers investigated a very important issue on metabolic disease treatment with Adenosine receptor chemical antagonist PSB-603 and compared the results with several relevant published papers in the recent literature on that field.

As the authors stated (line 285-288) it is not known “whether the reduction of body weight and the decrease in the amount of AT are due to stimulation or to the blockade of A2B receptors”. Thus, a comparison in the same experimental model, used in the present manuscript, with an agonist of A2B receptor (as BAY60-6583 cited by the authors, line 326-329) could have greatly increased the specific knowledge in the field and the innovative level of the research showed.

Moreover, several papers cited by the authors evidenced the important role of A2B receptor at the liver level and suggested some possible mechanisms at molecular level (line 341-349). The manuscript may have been improved and made more innovative if it had included a whole body metabolism assessment by clamp measurement and/or and histological and molecular characterization of liver morphological and functional changes during treatment.

Lastly, the low-grade inflammation characterizing obesity could be better evaluated by histological and IHC analysis performed at AT level than quantifying the circulating level of IL-6 and TNF-a.

Many questions remain open in the present manuscript, as correctly authors underlined (lines 285-288; 294-298; 359-363; 373-376; 395).

Minor Comments

Specify the time of the treatment (14 days) with PSB-603 in the Abstract (line 27)

Update the literature on the inflammation at AT level in the Introduction Section (lines 46-50)

Figure 2A is not sufficiently clear and the statistical analysis showed is redundant and confuse.

Line 114: is the sentence correct? PSB-603 reversed the increase in body weight during the time (14 days) of the treatment only.

Line 131 and everywhere in the text: the group of mice fed with High Fat-Diet and Sucrose for 14 weeks could be more precisely named HFDS-treated mice instead of obese mice.

The statistical analysis showed in Figure 3 and 6 should be comparing the treatments with the fat/sugar diet group instead of the standard diet (control) group.

Why author quantified only intraperitoneal AT? Many other AT deposits are studied and characterized in mice, such as subcutaneous inguinal AT and visceral perigonadal AT (see for reference Cinti S. Adipose Organ Development and Remodeling. Compr Physiol. 2018 Sep 14;8(4):1357-1431. doi: 10.1002/cphy.c170042). Moreover, due to the relevant papers in the literature also the Brown AT could be interesting to study upon the treatments.

Diagrams in Figure 5 are not so easy to understand and clear for the readers.

Lines 189-190: is the decrease of AUC upon theophylline-treatment significant if compared with fat/sugar diet group?

In Figure 7B, the fat/sugar diet group did not display a significant increase in the AUC during ITT, as generally observed in mice upon HFD. Please comment this result.

The effect of PSB-603 on cholesterol levels is not dose dependent (Figure 8) and the hypothesis suggested to explain this result is not so convincing (lines 359-360).

The Discussion start with a comment of Figure 4: maybe the authors could consider changing the order of the Figures accordingly to the progression of the Discussion.

In particular, the experiment testing the effect on the locomotor activity of the compound (PSB-603) was performed after a single administration while the treatment was done for 14 days (chronic studies). Moreover, no experiments were performed with the reference compound (theophylline) in the experimental setting used in the present study. Please comment these relevant decisions accordingly to the Discussion sentences (line 387-388)

Material and Methods (397)

4.3 : 0.35ml/kg = 350 microliters/g=10.5ml/mouse  for 1 ip injection?????????? (line 431)

Is “decapitation” of mice accepted by the ethics Committee in Poland? (line 482)

References have been corrected (lines 540, 554) and ordered by years (592-603)
